# Implicit GPS-based bicycle route choice model using clustering methods and a LSTM network

**Lucas Magnana**[1]*, **Herve Rivano**[1], **Nicolas Chiabaut**[2]

**1** CITI, INSA Lyon-Inria, Université de Lyon, Villeurbanne, France, **2** CITEC Ingénieurs Conseils SAS, Lyon, France

* lucas.magnana@inria.fr

**Data Availability Statement:** We used 2 datasets of GPS tracks. The first dataset (Veleval) is restricted because it can contain potentially identifying information. The restrictions have been imposed by INRIA's Research Ethics Committee. A

## Abstract

Biking is gaining in popularity all around the world as a healthy and environmentally friendly mode of transportation. Urban policies tend to encourage citizens to use bicycles. This can be done by creating new cycling infrastructures, the renovation of old ones or the deployment of bike-sharing systems (BSS). These policies having a cost, understanding and predicting the behavior of cyclists has become a necessity in order to optimize them. Classical methods analyzing cyclists' route choices use external factors and generated choice sets of paths along with a logit model to create a discrete route choice model. Nevertheless, few studies focus on the predictive capacity that this type of model can offer. In this paper, we developed a prediction-centered bicycle route choice model. Our model is created without using external factors or choice sets of paths as in the more classical methods. The idea of our method is to use deep and machine learning algorithms on GPS tracks. These algorithms learn representations from the data which replace explicit factors. To build the model, we clustered the GPS tracks using DBSCAN. The clusters allow to identify the cyclists' preferred road segments and are used to create paths using them. A method weighting the road graph weights is developed to create paths passing through the preferred road segments of a given cluster. A LSTM is finally trained in order to retrieve a cluster from a shortest path between an origin/destination pair. Tracks created by our model are more similar to the original GPS tracks than the shortest paths or tracks generated by a prominent path computation service.

## 1 Introduction

Urban mobility is experiencing a significant change with the return of the use of bicycles for daily trips. Motivations to leave motorized vehicles to ride a bicycle are numerous, such as health benefits [1, 2], money and time saving for individuals, and pollution reduction for cities [3]. Consequently, urban planners and policy makers have many incentives to promote bicycling as a transportation mode. Such a strategy requires redistributing urban space more equally between motorized vehicles and other transportation modes. How to deploy bike paths and bike lanes in order to create the optimal structure of bicycle infrastructures is becoming a trendy research topic. Indeed the number and the quality of bike facilities are positively related

data request can be sent to Mathieu Cunche, Associate Professor at INSA-Lyon / Inria, member of the Privatics team hosted by the CITI laboratory, at [mathieu.cunche@inria.fr]. The second dataset (Monresovelo) is accessible here: https://donnees. montreal.ca/ville-de-montreal/trajets-individuels-velo-enregistre-mon-resovelo. The other data used are publicly available through OpenStreeetMap.

**Funding:** This work has been partially supported by the French National Research Agency within the framework "Investissement d'Avenir", ref ANR-17-CONV-0004. Lucas Magnana thesis is funded by Ecole Urbaine de Lyon. No additional external funding was received for this study.

**Competing interests:** The authors have declared that no competing interests exist.

to the number of cyclists [4, 5]. To this end, a fine understanding of cyclist behaviors is crucial to deploy bike facilities efficiently. Predicting what paths a user may follow is decisive to anticipate on network's performance and constitutes a research challenge.

Thanks to the increasing number of bike-sharing systems (BSS), massive data of cyclists' journeys in cities are made available by municipalities. These data provide the origin station, the destination station, the travel time, and some information on the user for each trip. Studies using BSS data to analyze and predict bike flows are numerous [6–16]. One main objective of these studies is to facilitate the rebalancing of the system. The flows among stations are predicted by crossing traffic, weather, and social data in regression analysis, machine learning and/or deep learning algorithms. Some approaches involve clustering techniques to regroup stations that are geographically close and have similar check-in/check-out patterns [7, 14]. In 2019, Li et al [14] clustered stations according to their geographical positions and their transition matrices. They then used machine learning and multi-similarity-based inference models along with time, weather, and event-related data to predict the number of check-out at three different-scaled locations (entire city, clusters and stations). Finally, they designed a method using inter-cluster transition matrix and an approximation of the total number of check-out to infer the check-in number of each location. Their work outperform the prediction capabilities of precedent methods at the city and cluster level. However, they say almost nothing about the results at the station level, as their use is highly variable.

Neural networks have proven their effectiveness in prediction in recent years. Recurrent Neural Networks are NNs capable of handling time-dependent and sequential data. They have been widely used in fields using temporal data such as speech recognition, language modeling, translation or image captioning [17–19]. LSTM networks are RNNs specially designed to handle long sequences. Used on BSS data, LSTM networks outperform flows prediction's performances of classic statistical, deep and machine learning methods on station-level [11–13]. In 2019, Pan et al [13] used a LSTM network on data from Citi Bike System Data to predict the number of bikes rented and returned in the BSS system. They divided each day in 24 time steps and their model is capable of predicting the number of bikes rented and returned at each station at each time-step. The LSTM network outperforms the predictive capabilities of a classical neural network in that case, as the classical neural network does not take into account the sequential aspect of the data. However, studies studying flows in BSS systems only predict the behavior of BSS users, which may not be representative of the total number of cyclists. Moreover, the origin and destination points of BSS users are limited by the stations. It is probably a coarse and biased sampling of the space.

LSTM networks can also be used on dockless BSS systems that are gradually being implemented in the world. These systems offer the advantage of not restricting the origins and destinations of their users as bicycles can be picked up and dropped off anywhere in the city. Recent studies are dividing the city in several parts and use LSTM networks to predict the bike distribution [20] or the trip production and attraction in each one of the part [21].

Nevertheless, all the presented studies are based on origin/destination pairs, and do not include more information about the users' paths. Predicting O/D flows is not enough to model cyclists' behaviors. Even the studies predicting flows at a trip scale [9, 11–13] are focused on the origin, the destination and sometimes the time duration but do not consider the route followed by the cyclist, as the BSS data do not generally provide GPS tracks of trips. In contrast to car mobility studies, the reconstruction of paths followed by bikes requires more information than the origin, the destination and the trip duration. Bicycles have indeed more degrees of freedom to move than cars. In particular, route-choice models for cars are usually assuming they follow shortest paths [22], even if some studies put this hypothesis into perspective [23]. This assumption does not hold for cyclists [24]. Recently, cities have deployed bicycle counters

and made count data publicly available. They help to study cyclists' behavior in more detail than by using only origin/destination pairs. Some studies use these data to check the representativity of non public crowd-sourced GPS data [25], to identify environment factors influencing the bicycle volume [25, 26] or to create models that fit the pattern of seasonal bicycle demand [27]. Counting the number of cyclists can confirm/infirm the utility of a constructed facility and evaluate its usage rate. However, even when coupled with BSS data, these partial observations do not allow the reconstruction or prediction of the paths taken by cyclists without adding strong external assumptions. Methods for studying and predicting cycling behavior therefore rely on more detailed data than those provided by cities and BSS.

To the best of our knowledge, the only existing models able to predict cyclists behaviors at a trip scale are the discrete route choice models [24, 28–34]. These models require the characteristics of several observed paths used by cyclists, collected in the past through preferences surveys [28, 29], and more recently through GPS sensors [24, 30–34]. Choice sets of paths are generated to represent all the alternatives taken into account by the cyclists for each observed paths, and factors are chosen to describe these alternatives. A modified logit model is then used to assign a weight on each factor and calculate the probability of being chosen for each of the generated paths. A well-known issue associated with these models is that the set of all feasible paths is intractable and the actual choice sets of paths are unknown. As the model has to choose a route between the ones in a choice set, the accuracy of the model's behavior prediction is closely related to the quality of the sets. Moreover, choosing explicitly factors to describe the alternative paths allows only a partial analysis of route choices, as not all factors can be analyzed at the same time. This limitation is all the more present as the selected factors are practically the same in all studies. Designing new factors that can affect people's behavior is a difficult task involving multidisciplinary research. Today's factors are mainly infrastructural e.g. the number of turns or the proportion of bike facilities, global e.g. the length of the trip or the difference with the shortest path, dynamical e.g. weather or traffic volume, or social e.g. age, gender or car ownership. Finally, studies developing discrete route choice models mainly focus on the factors influencing cyclists' choices and understanding their importance, but seldom analyze their model's route prediction capability.

This paper aims to develop an implicit bike route choice model which use learning methods instead of external factors and do not require the generation of choice sets of paths. The objective of our model is to generate a cycling path when given an origin/destination pair. The model learns from previous individual observations to create paths that simulate cyclists' behaviors. The learning creates implicit representations of the data which replace the external factors used in the more classical methods. Our model uses learning tools classically used on BSS data on GPS data. This allows to keep the completeness of the GPS tracks and to make predictions of cycle tracks using only observed data, thus limiting the biases created by the generation of the choice sets of paths and the choice of external factors. The main contributions of our work can be summarized as follows:

- We identify cyclists' preferred roads by clustering GPS tracks passing through the same road segments.

- We develop a method to create a cycling path between a pair of O/D using a cluster of GPS tracks.

- We train a LSTM neural network to choose the cluster to use to create the most accurate cycling path from a pair of O/D.

- We compare the similarity between GPS tracks generated by cyclists and tracks created using several methods.

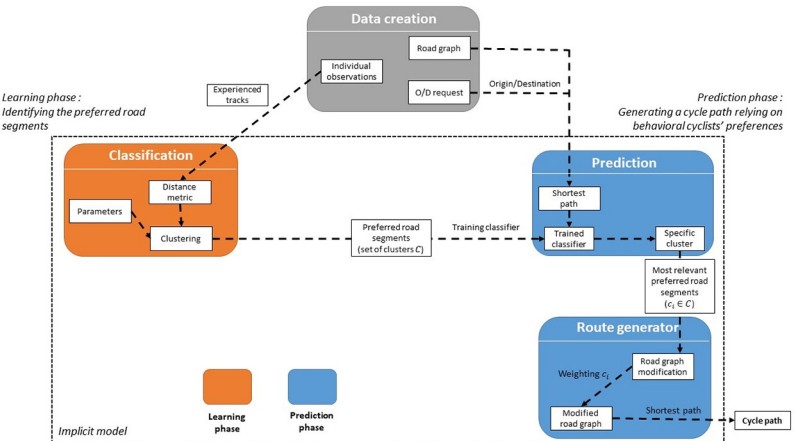

**Fig 1. Diagram summarizing our bicycle implicit route choice model.**

Fig 1 shows the structure of our method and model.

## 2 Data

Travel time (or cost) is one of the main determinants in the route choice of cars' drivers [22, 23]. While this factor remains crucial for cyclists, bike route selection is also influenced by other observations such as the safety of the roads used, the number of intersections, minimizing hills/turns, or riding dedicated facilities [35]. However, and particularly when it comes to commuting trips, cyclists are unwilling to make too many detours to optimize these factors because riding a bike requires a physical effort. Consequently, we assume that a cycle route is a trade-off between the shortest path and passages on preferred road sections. Making a qualitative and quantitative comparison between cyclists' routes and their equivalent shortest paths leads to a first intuition about the form and amount of detours that cyclists are willing to make. In this part, we first present the cyclists' tracks we use, then we detail how we generated shortest paths and finally we compare qualitatively and quantitatively the cyclists' routes and their equivalent shortest paths.

### 2.1 Experienced tracks

Despite the increasing number of GPS sensors and the number of studies using these kinds of data, few data-sets of biking GPS tracks are publicly available, and those are of small or medium size. Two data-sets are used in our study:

- A private set of GPS tracks coming from a multidisciplinary research project in University of Lyon called *Veleval*. Forty volunteer cyclists were asked to do commuting trips by bike from May 2016 to October 2019 in Lyon and Saint-Etienne, France. Each cyclist carried a GPS tracker during his trips. The result of this project is 2535 GPS tracks contained in GPX files. Each GPS track $t$ is a sequence of $N$ ordered points $t = \{p_1, \ldots, p_N\}$ with each point $p_t = (lat_t, lon_t, time_t)$ being made up of a latitude value, a longitude value and a timestamp.

- An open data-set provided by Montreal, Canada, collected by their *MonResoVelo* application (public). MonResoVelo is a smartphone application that cyclists could use while biking to be tracked during their trips. Data collected by the city were then used to improve the development of cycling facilities. 4881 GPS tracks from this application are public. They have been

generated by cyclists from June 2013 to October 2015 and are contained in a JSON file. Each track $t$ is a sequence of $N$ ordered points $t = \{p_1, \ldots, p_N\}$ with each point $p_t = (lat_t, lon_t)$ being made up of a latitude value and a longitude value. Other information such as the departure and arrival time are available but we only used the GPS points. The tracks are already map-matched and anonymized.

We decided to ignore the temporal properties of the tracks and to only focus on the spatial ones, because it makes the setting of the learning algorithms used later easier. We thus removed the timestamps from the *Veleval* project tracks. We used the Ramer-Douglas-Peucker algorithm on both datasets to decrease the number of GPS points per tracks. Fewer GPS points means shorter processing time. For the sake of brevity, only the Veleval data are presented in the paper. We preferred the Veleval dataset to the MonResoVelo one because it contains almost exclusively commuting trips. As our model has to create paths using an O/D pair, using a dataset of cyclists with a specific destination for each of their trips seemed more relevant. The purpose of tracks in MonResoVelo dataset can be among other things the practice of sports, or not even be defined. The results for MonResoVelo are nevertheless showed in S1 Appendix. For the rest of this study, the cyclists' GPS tracks will be referred to as experienced tracks.

## 2.2 Computed tracks

Because we want to compare the experienced tracks to shortest paths, the set of shortest paths between any origin/destination pairs has been computed with a Dijkstra algorithm. The algorithm was executed on a road graph $G$ generated with OpenStreetMap. A road graph is a weighted directed graph which is a pair $G = (V, E)$ with $V$ a set of vertices and $E \subset \{(x, y)|(x, y) \in V^2 \land x \neq y\}$ a set of edges which are ordered pairs of vertices. Each edge has a weight assigned to it. In a road graph, vertices represent the intersections and edges represent the road segments linking two intersections. We generated the road graphs of the city of Lyon and its surroundings as well as the city of Saint-Etienne and its surroundings. Their sizes are respectively 190276 edges connecting 68380 nodes and 21314 edges connecting 7559 nodes. The weights assigned to the edges are the lengths of the road segments they represent. Dijkstra algorithm uses the weights of the edges to compute the shortest path between two points. Fig 2 shows the distribution of distance differences between the experienced tracks and the generated shortest paths. The dashed red line shows the first experienced track longer than its shortest path. We can see that 10% of the experienced tracks are shorter than their shortest paths. On these 10%, there is 4.5% of tracks that are less than 100m shorter. This is mainly due to the impreciseness

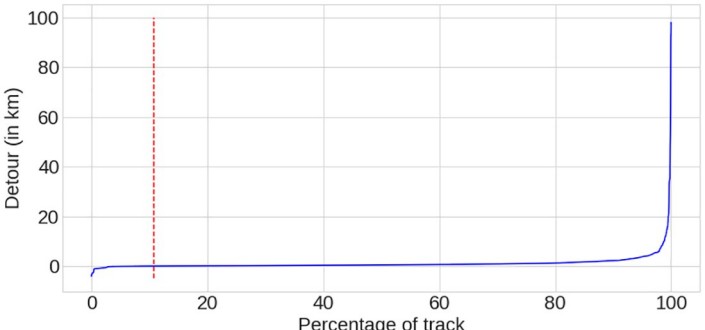

**Fig 2. Distribution of distance differences between the experienced tracks and the generated shortest paths (sorted).**

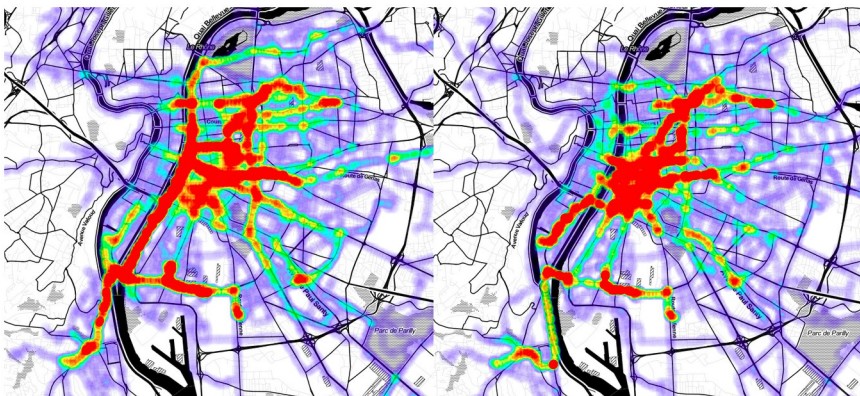

**Fig 3. Heat-maps of the experienced tracks (on the left) and the computed tracks (on the right).**

of the GPS trackers wore by the cyclists: if the track and the shortest path pass by the same way, their distances will not be the same. Sometimes the track will be shorter than the shortest path generated. For the remaining 5.5%, the difference between the tracks and the shortest paths are between 100m and 4km. The cause of this phenomenon is the generated road graph which is not fully accurate. Some roads used by cyclists are not in the graph, resulting in a detour made by the Dijkstra algorithm compared to the observed track. For the rest of this study, the shortest paths will be referred to as computed tracks.

## 2.3 Comparisons between experienced and computed tracks

**2.3.1 Qualitative comparison.** To verify that the cyclists' tracks are different from the shortest paths between their origins and destinations, we generated two heat-maps. The first one contains all the experienced tracks, and the other contains all the computed tracks. We show these heat-maps on Fig 3. In the city center, the experienced tracks' activity is focused on specific roads, while computed tracks are using roads uniformly to minimize the distance traveled. As we expected, cyclists tend to use roads that they find more suitable for cycling. This is particularly visible with the left banks of the Rhône river which are closed to cars and have been specially arranged for pedestrians and bicycles, making them a privileged path for Lyon's cyclists. On the other hand, the experienced tracks' activity is distributed more evenly on the roads in the periphery, preventing the visual distinction of the most used roads. This is consistent with our hypothesis: cyclists are willing to use preferred roads, even if it means not using the shortest path between their origins and destinations. However, they seem to want to minimize the distance traveled to access these preferred roads. That's why we don't see preferred roads in the periphery. The cyclists' routes would thus be a trade-off between specific road segments and shortest paths to access them.

**2.3.2 Quantitative comparison.** Cyclists in our data-set tend to use preferred road sections even if it means going away from the shortest path to their destinations. However, making a detour has a physical and a time cost. This cost must not cancel the benefits of using the preferred routes. Knowing that, cyclists must have a threshold of acceptance beyond which they refuse to make a larger detour. To verify this hypothesis, we computed the difference between the distances of the experienced track and the distances of their equivalent computed tracks. We plotted a cumulative distribution function of them (Fig 4). The maximum detour reaches almost 100km (see Fig 2), but the CDF shows that 92% of detours are less than 2.5km. This validates our hypothesis that most cyclists do not want to make detours greater than a

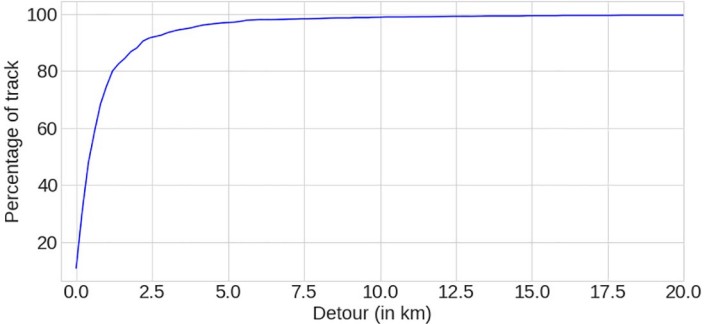

**Fig 4. CDF of the differences of distances between experienced and computed tracks.**

certain distance. On our data-set, this distance is 2.5km. As we only wanted to study commuting trips in this paper, we decided to remove the 8% of tracks with a detour greater than this distance. These 8% corresponded to 233 tracks, which means that our data-set contains 2301 tracks after this step.

## 3 Preferred road segments identification

Cyclists in our dataset tend to make detours to use preferred road segments. Paths created by our model must therefore make the same detours to simulate cyclists' behavior. We need to first identify the preferred road segments. Using the entire data-set allows to identify the preferred road segments used by most cyclists, but not to find those used by only a part of them. We assume that decomposing the data-set will allow us to access more detailed information and identify all preferred road segments, regardless of the number of cyclists using them. Therefore we divide our data-set of experienced tracks into clusters (see the *Classification* block on Fig 1). Each cluster is grouping experienced tracks that share one or several road segments. Road segments identified in the clusters are then used as preferred road segments to simulate detours made by cyclists. In this section, we first define the similarity metric used to generate the clusters, then we argue on the choice of the clustering algorithm, and finally explain how we set-up it.

### 3.1 Distance metric

Our clustering algorithm aims to group experienced tracks passing through one or several common road segments. We need to quantify the proportion of common road segments between two tracks for our clustering algorithm to achieve this goal. Several methods exist to do it, the most known being using a map-matching algorithm. However, we decided to use a simpler and easier method, as developing a map-matching algorithm is difficult. Moreover we don't need all the information it would provide. We used a space reduction to identify the proportion of common road segments two tracks share. The map is divided into cells of 55m by 38m. For time and resource reasons, we only computed cells intersecting at least one track. Each track has a direct set of cells through which it passes. With this size of cell, it happens that two tracks on the same road passed through two neighboring cells instead of a unique one, because of the GPS imprecision. To correct this, after creating all the direct sets, each track is assigned an extended set of cells $\tau$ that contains the cells through which it directly passes plus all their computed neighboring cells. The extended sets therefore depend on the tracks that we

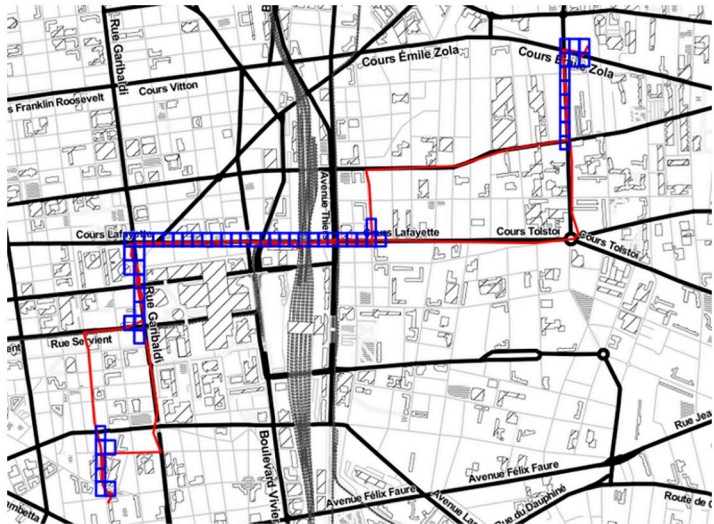

**Fig 5. Graphical example of the intersection between the extended sets of cells of two tracks.**

computed the direct sets for. Fig 5 shows a graphical example of the intersection between the extended sets of two tracks.

With $t_1$ and $t_2$ two tracks, $\tau_1$ the extended set of $t_1$ and $\tau_2$ the extended set of $t_2$, the distance $D_{t_1 t_2}$ between $t_1$ and $t_2$ is the Jaccard distance [36] between $\tau_1$ and $\tau_2$, defined as:

$$1 - \frac{|\tau_1 \cap \tau_2|}{|\tau_1 \cup \tau_2|}$$

Once the direct sets of the 2301 experienced tracks have been computed, we calculated their extended sets and created the distance matrix of our data-set.

## 3.2 Clustering algorithm

The relevance of using a clustering algorithm in a situation depends on the type of data (noisy or not, big or small data-set. . .) and the properties of the searched clusters (number, size, shape. . .) [37]. We want to create clusters to identify preferred road segments. To enable this identification, each track in the same cluster must share a certain amount of road segments with all other tracks. In Section 3.1, we developed a distance metric to quantify how much two tracks do or do not share road segments. Our clustering must therefore create groups of tracks that are dense with respect to our distance metric. This is the goal of density-based clustering algorithms, and in particular of DBSCAN, the most widely used of them.

DBSCAN (Density-Based Spatial Clustering of Applications with Noise) is a density-based clustering algorithm that is able to find arbitrary shaped clusters and is well-suited to data with noise. Given a set of points and a distance, DBSCAN will group points closely packed together and mark as outliers points that have not a sufficient number of close neighbors. It does not require a predefined number of clusters, and has two parameters. We explain their meanings and how we set them up in Section 3.3. DBSCAN's properties make it a clustering algorithm adapted to our case. It generates groups of tracks that all share a certain amount of road segments which can then be used as preferred road segments. It also discards tracks that do not share enough road segments with others, allowing our model to use only tracks that represent minimally widespread behavior in our dataset.

To confirm our intuition, we compared DBSCAN with k-means, the most known clustering algorithm. K-means is not density-based, and requires a number of clusters. It creates clusters of similar shapes and sizes, without taking noise into account. As we expected, some clusters created by k-means were large and contained tracks with no common road segments. In comparison, clusters created with DBSCAN all contain at least one preferred road segments visually noticeable.

## 3.3 Parameters

As said earlier, DBSCAN requires 2 parameters: *MinPts* and $\varepsilon$. *MinPts* defines the minimum number of points in a dense region to consider it a cluster. $\varepsilon$ defines the maximal distance two nearby points can have to belong to the same region. Calibrating DBSCAN is always a tedious task. To determine the optimal parameters, we first have to define variables that are consistent with the shape of clusters we aim to find. Indeed, we search for clusters that respect these properties:

- **Noise:** A badly parameterized DBSCAN can either consider a majority of the data as noise, or not detect any noise at all. Suppose a majority of the experienced tracks are considered to be noise. In that case only a little part of our data-set will be clustered resulting on preferred road segments representing only a small portion of the cyclists in our data-set. On the other hand, if no tracks are considered noise, the clusters will contain tracks that do not share common road segments resulting in poorly defined preferred road segments.

- **Number of clusters:** Too many clusters means that they do not contain on average enough experienced tracks to identify the preferred road segments used by many cyclists. Conversely, few clusters means that they contain on average too many experienced tracks. As a consequence, only the most used preferred road segments will be identified.

- **Number of large clusters:** We defined a large cluster as a cluster that contains more than 1.5% of our data-set. Maximizing the number of large clusters ensures that the preferred road segments used by a large number of cyclists are identified.

- **Silhouette coefficient:** Silhouette is a coefficient used to validate the relevance of clusters. For each clustered point, silhouette measures how similar the point is to its cluster compared to other ones. Silhouette ranges from -1 to 1, a high value indicates that the point is well matched to his cluster, and far enough to neighbor clusters. We computed the mean of the silhouette of all points clustered. A good silhouette score means that the preferred road segments are well defined in our clusters, and that none of them are identifiable in several clusters

Fig 6 shows the evolution of these four variables with respect to $\varepsilon$ while *MinPts* is set to 3. We conducted the following analysis with several *MinPts* and 3 seems to be the value giving us the best results. A low value of $\varepsilon$ means that only points very close to each other will be clustered together. We can see that the noise decreases as the value of $\varepsilon$ increase because the more distant points can be put in the same cluster, the less DBSCAN finds points that cannot be included in any cluster. The number of clusters, the number of large clusters and the silhouette score begin by increasing until they reach a maximum point after which they decrease. As $\varepsilon$ is increasing, clusters are made of points that are more and more distant from each other. As a consequence DBSCAN creates more and bigger clusters. However after a certain value of $\varepsilon$, the clusters will merge together decreasing their number. As the merges go by, the large clusters will begin merging with each other, reducing their number until there are few clusters left.

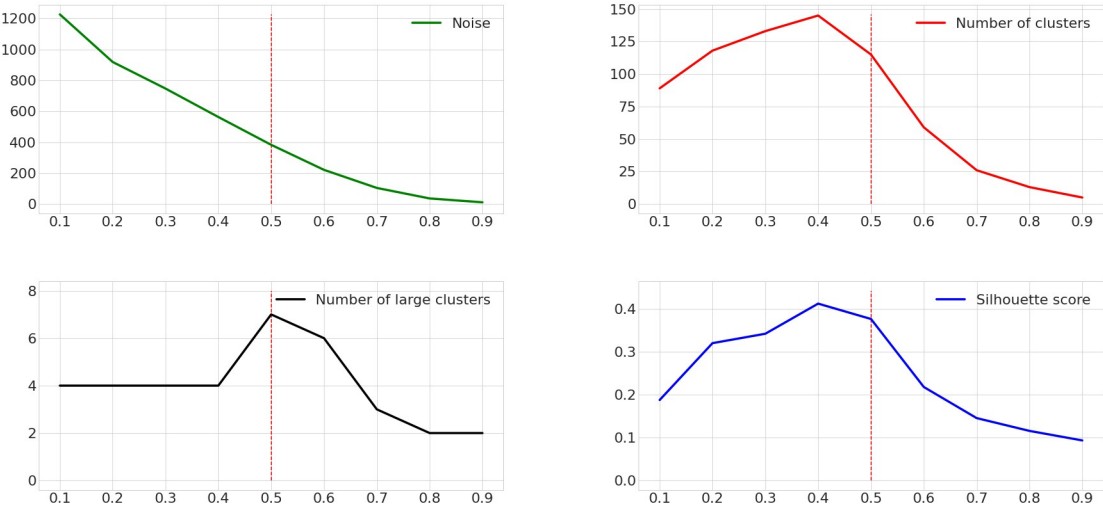

**Fig 6. Evolution of the noise (green line), the number of clusters (red line), the number of large clusters (black line) and the mean coefficient silhouette (blue line) with respect to ε and *MinPts* = 3.**

This process also decreases silhouette coefficient value because when clusters merge, points far enough to be in two different clusters end up in the same one.

In order to prevent large clusters from having experienced tracks that do not share any road segments and therefore to have poorly defined preferred road segments, we displayed all of them as heat-maps. Helped by the graphs and the heat-maps, we identify *MinPts* = 3 and ε = 0.5 as relevant parameters. We can see on Fig 6 that with this set up we have 115 clusters, we maximize the number of big clusters with seven ones, the mean silhouette coefficient is 0.37 and there are 383 outliers which represents 16.6% of the tracks.

The heat-maps of the largest cluster found by DBSCAN in Lyon with our set up is shown in Fig 7. It contains 140 experienced tracks. The left heat-map shows them. The right one shows their corresponding computed tracks. As we expected, the main preferred road segments showed by this cluster are concentrated on the left bank of the Rhône, one of the most popular cycling facilities in Lyon. All the experienced tracks in the cluster pass through at least one of

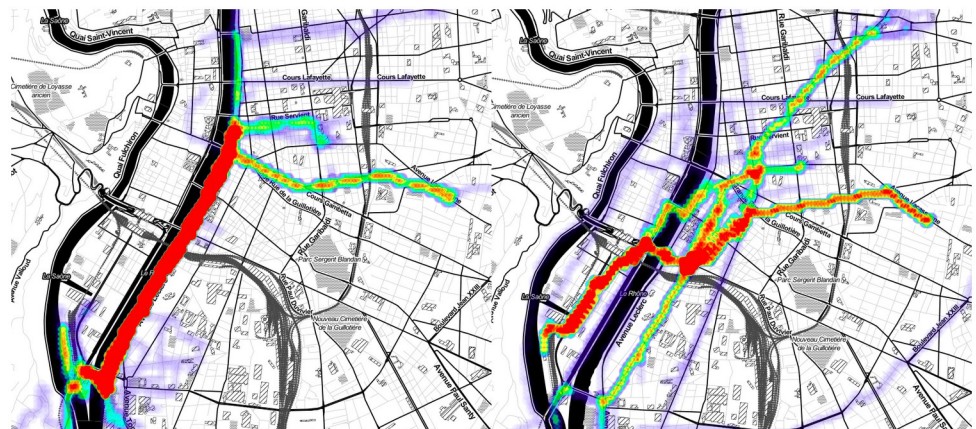

**Fig 7. Heat-maps representing the experienced tracks contained in the largest cluster found by DBSCAN in Lyon (left) and their equivalent computed tracks (right).**

these bank's road segments, and about 95% of the former pass through several of the latter. In comparison, less than 35% of the corresponding computed tracks pass through at least one of the bank's road segments, and less than 5% of the former pass through several of the latter.

## 4 Behavioral path generator

Once our clusters created, we used them to compute experience-based tracks. Using a cluster of experienced tracks and an origin/destination pair, our goal is to create a path that pass through the preferred road segments identified in the cluster (see the *Route generator* block on Fig 1). However, the paths must not straying too far from the shortest path so as not to lengthen the distance to an unacceptable extent. Our approach relies on the Dijkstra algorithm with a weighted road graph.

### 4.1 Cyclability coefficient

Each cell c has a set $\delta_c$ of tracks passing directly through them and a set $v_c$ of tracks passing through their direct neighbors. Once all cells have been computed, we can calculate for each one of them a normalized cyclability coefficient $\mathcal{C}_c$ which is defined as:

$$\mathcal{C}_c = \frac{|\delta_c \cup v_c|}{\max_m |\delta_m \cup v_m|}$$

Thus, the cyclability coefficient takes values in [0, 1], as the cell *m* has a coefficient of 1. This coefficient can be defined and computed considering the tracks of the whole data-set or only the tracks of a specific cluster. Note that the heat-maps shown in the paper are colored following the cyclability coefficient. We used this coefficient to modify the road graph as follows.

### 4.2 Road graph modification

Given a cluster c, we defined an algorithm to modify the weights of the edges of a road graph G (Algorithm 1). *computeCells* takes one or several routes as input and returns the extended set of cells through which these routes pass. *e* is an edge of the road graph. It contains two GPS points that represent the position of the two intersections at its end and a weight. The algorithm modifies the weight of *e* according to the mean cyclability of the cells intersecting *e*. Here the cyclability is computed with respect to the chosen cluster. With this algorithm, the road segments that are the most used by the given cluster have a lower weight. Dijkstra's algorithm is hence more likely to select them when calculating the shortest-path between o and d. Browsing each edge of the road graph is time consuming, but it needs to be computed only once per cluster.

**Algorithm 1:** Road graph modification wrt a cluster

```
Data: A cluster c
Result: The road graph with modified weights
τ_c ← computeCells(c)
/* τ_c contains the set of cells through which routes in c pass       */
for e ∈ G do
  /* for each edge e in the road graph G       */
  τ_e ← computeCells(e);
  /* τ_e contains the set of cells through which e pass       */
  τ ← τ_c ∩ τ_e;
  m ← mean(C_c, c ∈ τ);
  /* C_c is the cyclability coefficient of the cell c       */
  e.w ← e.w*(1 − m)
  /* e.w is the weight of e       */
end
```

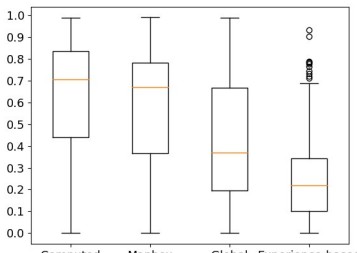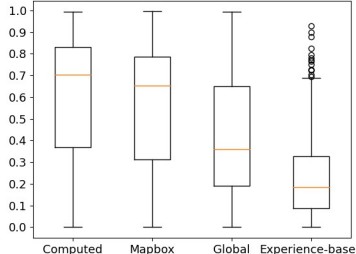

**Fig 8. Box-plots of the distance between the experienced tracks and the four other types of tracks.**

## 4.3 Quality of the experienced-based tracks

We computed an experience-based track between the origin and the destination of each experienced track. We used an oracle to choose the cluster to use to modify the road graph. For each experienced track, the oracle knows from its O/D pair on which cluster it belongs. Section 5 is dedicated introducing a practical implementation replacing this oracle.

We calculated the distance between the generated experience-based track and the original experienced track using our distance metric. The smaller the distance, the better the experience-based tracks simulate the cyclists' behavior. A bias of this method is that the experienced tracks that we are trying to simulate are directly contained in the clusters used to modify the road graph. For clusters that do not contain many tracks, the modification of the road graph could be very influenced by the track that we want to simulate. To overcome this issue, 20% of randomly chosen experienced tracks are removed from the clusters. We created the experience-based tracks for all the experienced tracks. We compared the results between the 20% of removed ones and the remaining 80% in Fig 8. To evaluate the quality of our experience-based tracks, we compared them with three other types of tracks:

- **Computed tracks:** These tracks are the shortest paths computed with the Dijkstra algorithm. The comparison highlights the detours performed by the cyclists.

- **Global experience-based tracks:** We created a modified graph using the cyclability coefficient computed on the whole data-set. We executed Dijkstra's algorithm on it to create global experience-based tracks. It enables us to evaluate the relevance of using clusters separately instead of using all tracks at once to modify the road graph. For now, the global experienced-based tracks are reffered as global tracks.

- **Mapbox tracks:** In addition we compare our protocol to a prominent path computation service of the market, Mapbox. Inputs of this commercial service are an origin/destination pair. Many other calculators exist but we decide to focus on this one for sake of simplicity. These tracks allow us to compare our experience-based tracks with tracks generated by a commercial service used by cyclists to know which path to use.

The Fig 8 shows two box-plots of the distance between the experienced tracks and the four other types of tracks. The experienced tracks are separated in two: the left graph shows the results with the 20% of experienced tracks removed from the clusters and the right one shows the results with the remaining 80%.

The results with the removed experienced tracks and the experienced tracks remaining in the clusters are similar. This shows that the quality of the experience-based tracks generated with the clusters is the same whether they contain the experienced tracks or not. We are only

interested in the results of the removed experienced tracks (the left box-plot on Fig 8), because we want our model to work with tracks that were not used to create it.

The furthest tracks from the experienced tracks are the one computed by the Dijkstra algorithm on the OpenStreetMap road graph. They are followed by the Mapbox tracks. This is not surprising since this service uses a (non-disclosed) shortest path algorithm or heuristic on their own road graph, to the best of our knowledge. The global tracks do better than the two previous types. The value of the median distance goes from about 0.65 for Mapbox tracks to about 0.35 for global tracks. However, the interval between median quartiles (i.e. the interval in which the middle half of the distances are found) is 0.41 wide, from 0.22 to 0.63. In addition, a significant number of global tracks have no common road segments with the experienced tracks they try to reproduce (those with a distance of 1). One possible explanation is that our algorithm prioritizes the road segments used by a significant number of experienced tracks. Our data-set is however not equally distributed geographically: some areas are less used than others (see Fig 3). Tracks using road segments in these lesser represented areas are therefore poorly deviated by our weight modifications. This further endorses the validity of our strategy of making cluster-wise modifications instead. Finally, the experience-based tracks computed by our method are the closest to the experienced tracks with a median distance at about 0.2. Moreover the distributions of the distances is the narrower. The median quartiles are only 0.25 apart and few outliers have a distance greater than 0.7. No experience-based track have a distance of 1.

Using the clusters allows the creation of tracks that are more similar to experienced tracks than shortest paths, Mapbox tracks or tracks generated using all the data-set at once. However, we need to implement a method to associate an O/D pair with a cluster.

## 5 Classifier

Experience-based tracks generated with our method managed to approximate the behavior of cyclists from our data-set. They are more similar to the experienced tracks than the tracks used in comparison. Nevertheless, until now, our method needs an oracle that knows the cluster to use given an O/D pair. To replace it, we trained a classifier using our computed tracks (see the *Prediction* block on Fig 1). For any computed track, its goal is to retrieve the cluster on which the corresponding experienced track belongs. We used cross-validation to ensure that the classifier has a good accuracy with tracks not used during his training. Once trained, we can use it to create an experience-based track from any origin to any destination in the road graph. To do so, we take an origin and a destination point and execute Dijkstra's algorithm to create a shortest path. We send the shortest path to the trained classifier which returns a cluster. We finally use the cluster as explained in Section 4.

### 5.1 LSTM Neural Networks

A computed track is a set of points that follow each other in a particular order. We must therefore choose a classifier capable of handling sequential data. Recurrent Neural Networks (RNNs) are a type of classifier which have recently shown good results on this type of data. In addition to its input data, a RNN has a hidden state $h_t$ which contains information about the data it previously processed. This allows it to take into account the temporal dependencies of the data it processes. We first tried to use a classical RNN for classifier. But it did not give us conclusive results, as some of our computed tracks were too long. Indeed, RNNs are confronted with the vanishing gradient problem which prevents them from learning long-term dependencies [38, 39]. The Long Short-Term Memory Neural Networks [40] (LSTMs) have been developed to overcome this issue. LSTMs are a particular type of RNN which include a

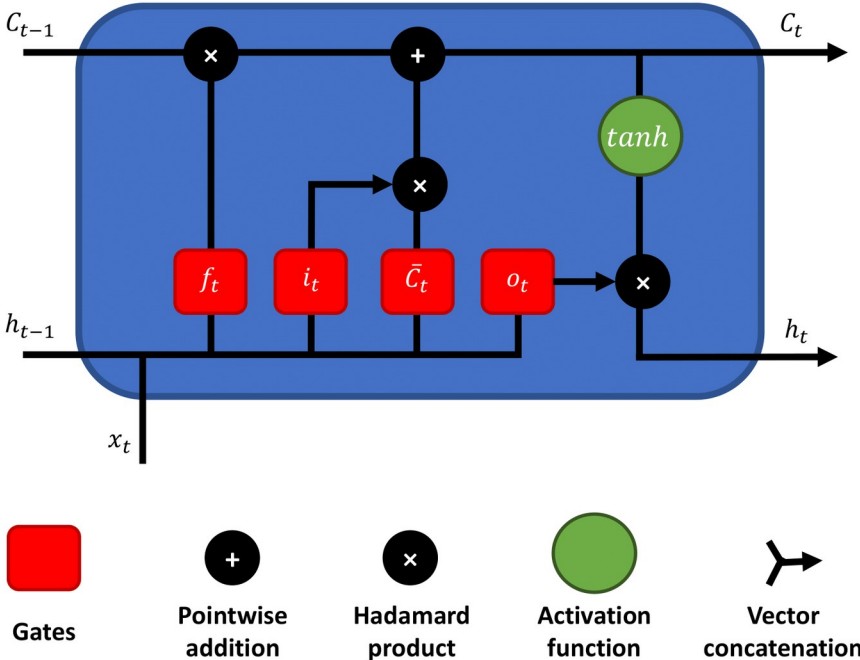

**Fig 9. Diagram showing the structure of a LSTM cell.**

cell state in addition to the hidden state. The purpose of the cell state is to keep the information about the long term dependencies of the data. Fig 9 is a diagram showing a LSTM cell. LSTM networks have shown better results than classical RNNs as computed tracks classifiers. We therefore chose this type of classifier for our final route choice model.

The information contained in the cell state $C_t$ is controlled and modified by three gates. First, the forget gate $f_t$ deletes the useless values from the cell state. Then, the input gate decides what new values will be added to it. The input gate is divided in two parts. The first part $i_t$ decides which values will be updated, the second part $\tilde{C}_t$ generates candidate values that can be added to the cell state. Finally, the output gate $o_t$ decides which values of the cell state is used to create the output $h_t$. The three gates are defined as followed:

$$f_t = \sigma(W_f.[h_{t-1}, x_t] + b_f)$$

$$i_t = \sigma(W_i.[h_{t-1}, x_t] + b_i)$$

$$\tilde{C}_t = tanh(W_C.[h_{t-1}, x_t] + b_C)$$

$$o_t = \sigma(W_o.[h_{t-1}, x_t] + b_o)$$

$x_t$ is the element of the input sequence $x$ at timestamp $t$ and $h_{t-1}$ is the hidden state computed by the LSTM at the previous timestamp ($t-1$). $h_1$ is randomly initialized. $W_f$, $W_i$, $W_C$, $W_o$ and $b_f$, $b_i$, $b_C$, $b_o$ are respectively the weight matrices and the bias vectors of the different gates. The cell state $C_t$ and hidden state $h_t$ at the current timestamp $t$ are calculated using the results of

the different gates:

$$C_t = f_t * C_{t-1} + i_t * \tilde{C}_t$$

$$h_t = o_t * tanh(C_t)$$

with the operator '*' being the Hadamard product and $C_{t-1}$ the cell state computed by the LSTM at the previous timestamp ($t-1$). $C_1$ is randomly initialized. $\sigma$ and *tanh* are non-linear activation functions defined as:

$$\sigma = \frac{1}{1 + e^{-x}}$$

$$tanh = \frac{e^x - e^{-x}}{e^x + e^{-x}}$$

When the $N$ elements of the input sequence $x = \{x_1, \ldots, x_{t-1}, x_t, \ldots, x_N\}$ have been passed to the network, the last hidden state $h_N$ is used for the classification.

## 5.2 Tracks pre-processing

A computed track is a sequence of $N$ ordered points $t = \{p_1, \ldots, p_N\}$ with each point $p_t = (lat_t, lon_t)$ containing a latitude and a longitude value. LSTM networks are well-suited for sequences, but continuous values of longitude and latitude points need to be discretized for the LSTM to converge. We took as an example the work of Crivellari et al [41] who used a LSTM network to predict the next steps on touristic paths. They transformed GPS tracks into sequences of touristic zones before passing them in a LSTM. As we could not use bike paths as bicycle zones because not all experienced tracks pass on them, we created our own ones. We clustered the cells generated for the similarity metric using a k-means algorithm, their geographical positions and their cyclability coefficients. The result is 1577 bicycle zones geographically close and having the same bike usage. We used the computed tracks' direct sets of cells and transformed them into sequences of bicycle zones using the cluster number of cells in the sets. The inputs of the LSTM are sequences of integer representing the bicycle zones numbers.

## 5.3 Set-up and performances

Our NN is made of a two layers LSTM followed by a single-layer perceptron. It is trained with the Adam optimizer and at a learning rate of 0.0005. During the training process which lasts 8500 steps, the input sequences are sent to the NN in batches of 30. The NN's output is a list of 115 log-probabilities obtained by using $log(Softmax(x))$. These log-probabilities correspond to the probability that the experienced track used to create the input computed track belongs to each cluster. The negative log likelihood loss is calculated and back-propagated. The computed tracks are divided into two sets: the training set which contains 80% of the tracks and the testing set which contains the remaining 20%. We took the 20% of experienced tracks that we removed from the clusters in Section 4.3 to create the testing data-set. During the training phase, only tracks belonging to the training set are sent. We used tracks belonging to the testing set to see how successful our NN is at classifying tracks that were not used during its training. Our trained LSTM has a success rate of 97% on the training set and 79.1% on the testing set.

Fig 10 shows the good and bad predictions of the LSTM in each cluster in the testing data-set. There are few clusters for which the LSTM makes only false predictions, and these clusters are all very small. It is possible that these clusters being small, they are not represented during

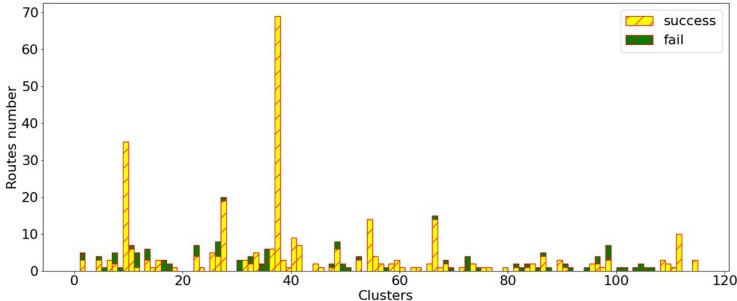

**Fig 10. Successed and failed predictions of our LSTM in each cluster in the testing data-set.**

the training phase of the LSTM. To overcome this issue, we tried to create our testing data-set by taking 20% of the experienced tracks of each cluster instead of taking 20% of the whole data-set, without success.

To evaluate the performances of our final model, we conducted the same experiment as in Section 4.3. We created an experience-based track for each experienced track, but this time we used our LSTM instead of an oracle to choose the cluster to modify the road graph. We then calculated the distance between them and the experienced tracks. We compare this distance with the one between the experienced tracks and the other types of experience-based tracks (i.e. the global ones and the one created with the oracle). Once again, we differentiate the results obtained with the training data-set from those obtained with the testing data-set, and show them in Fig 11.

With the training data-set, the results of the LSTM and the oracle are almost the same. We note a more significant number of outliers with a distance of more than 0.7 created with the LSTM. The LSTM having a 97% success rate in the prediction of the clusters with the training data-set, the clusters used by the oracle and the LSTM are the same almost all the time.

For the testing data-set, the median of the distances of the tracks generated with the LSTM is almost identical to the one generated with the oracle. However, more tracks have a distance greater than 0.7, and these tracks are no longer considered outliers. Moreover, the interval of the median quartiles goes from 0.1 to 0.35 for the oracle to 0.13 to 0.48 for the LSTM. This shows that for our method to create a cycle path simulating well the behavior of cyclists, using the good cluster to modify the road graph is essential.

The Fig 12 shows four examples of experience-based tracks created with our model, along with the original experienced tracks and the corresponding computed tracks for each one of them.

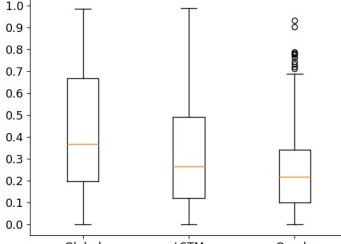
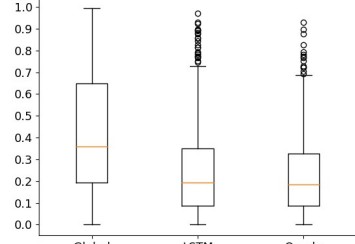

**Fig 11. Boxplots of the distance between the experienced tracks and the experience-based tracks.** The left one shows the testing data-set and the right one the training data-set.

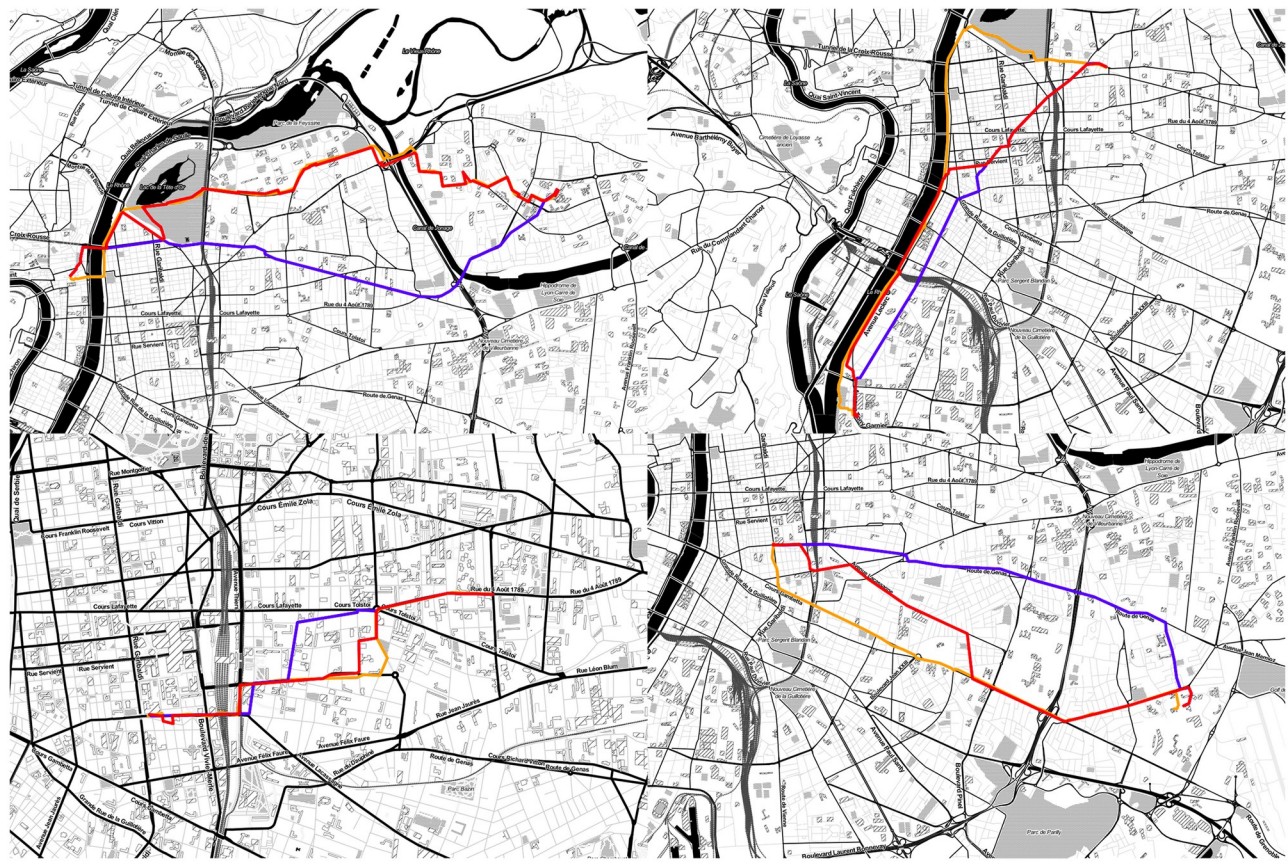

**Fig 12. Maps showing four experienced tracks (orange lines), the corresponding computed tracks (blue lines) and the experience-based tracks created with our model (red lines) between the experienced tracks' O/D pairs.**

## 6 Conclusion

Understanding cyclists' behavior and being able to predict it is important for creating relevant cycling facilities. This paper proposed a data-driven implicit bicycle route choice model which does not use pre-determined explicit factors or route coice sets of paths. This model is built on GPS tracks generated by cyclists in Lyon and Saint-Etienne, and uses the DBSCAN clustering algorithm to group similar tracks. The cyclists' preferred road segments are identified based on these groups, and then used to create tracks simulating their behavior. To choose the cluster containing the most relevant road segments to use, we computed the shortest path between the origin and the destination of all the GPS tracks and trained a LSTM to find the cluster to which the GPS track belongs from its shortest path. Our final model relies on four steps:

- With an origin/destination pair and an unmodified road graph, a shortest path is created using Dijkstra's algorithm.

- The shortest-path is sent to the LSTM which returns a cluster number.

- Tracks in the cluster are used to modify the weights of the edges of the road graph.

- Dijkstra's algorithm is used on the modified road graph to create a track that simulates cyclists' behavior.

The tracks created by our model are more similar to the GPS tracks than the shortest paths. Moreover, we compared our generated tracks to tracks given by Mapbox, a well-known routing API that allows the routing to be executed for cyclists. The similarity between the Mapbox's tracks and the GPS tracks is much lower than the one between the GPS tracks and the tracks generated by our model. Finally, to evaluate the relevance of using clusters to modify the road graph instead of using all the GPS tracks at once, we created a modified road graph using all the GPS tracks. The tracks generated with this road graph are less similar to the GPS tracks than the tracks generated with our clusters.

## 7 Discussion

A first limitation of our work is the dataset we used. We were unable to find a massive public dataset of GPS tracks generated by cyclists. Therefore, we used a private one that we already owned. It gathers GPS tracks of a small group of cyclists who mainly made commuting trips. It is thus made up of tracks that are similar to each other, facilitating the clustering work. In future research, it would be meaningful to use a set of GPS tracks with more data and generated by more cyclists. In the S1 Appendix, we used the data-set *MonResoVelo* which contains almost 5000 crowd-sourced GPS tracks, and created a model using the same methodology. The silhouette score of the clusters dropped to almost 0, and the performances of the LSTM dropped to 55%. However, even with a silhouette score at 0, the tracks created with the clusters are still way more similar to the GPS tracks than the three other types of tracks used as a comparison. The silhouette score seems to influence the LSTM's performances but not much the quality of the identified preferred road segments. We assume that the preferred road segments identified by several clusters are the same, dropping the silhouette score and preventing the LSTM from finding the right cluster given a computed track, but that these preferred road segments are well defined and allow the creation of tracks simulating well the behavior of cyclists. Using another clustering algorithm could allow a better silhouette score, or future classifiers could have better performances with a low silhouette score.

Secondly, our method does not explain the causes of the cyclists' choices and preferences it identifies. These identifications depend on the representations learned by the learning algorithms which are by nature implicit.

Finally, we did not use the spatial properties of the GPS tracks, as we wanted to focus only on the spatial ones. In future works, using temporal properties could improve the performances of the learning algorithms and therefore the accuracy of the generated cycling paths.

## Supporting information

**S1 Appendix.**
(PDF)

## Author Contributions

**Writing – original draft:** Lucas Magnana, Nicolas Chiabaut.

**Writing – review & editing:** Herve Rivano.

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
