## [Decision Letter · Decision Letter 0]

16 Jul 2021

PONE-D-21-18342

Implicit GPS-based bicycle route choice model using clustering methods and a LSTM network

PLOS ONE

Dear Dr. Magnana,

Thank you for submitting your manuscript to PLOS ONE. After careful consideration, we feel that it has merit but does not fully meet PLOS ONE’s publication criteria as it currently stands. Therefore, we invite you to submit a revised version of the manuscript that addresses the points raised during the review process.

We look forward to receiving your revised manuscript.

Kind regards,

Jinjun Tang

Academic Editor

PLOS ONE

Reviewers' comments:

Reviewer's Responses to Questions

**Comments to the Author**

1. Is the manuscript technically sound, and do the data support the conclusions?

Reviewer #1: Partly

Reviewer #2: Yes

2. Has the statistical analysis been performed appropriately and rigorously? 

Reviewer #1: N/A

Reviewer #2: Yes

3. Have the authors made all data underlying the findings in their manuscript fully available?

Reviewer #1: No

Reviewer #2: Yes

4. Is the manuscript presented in an intelligible fashion and written in standard English?

Reviewer #1: Yes

Reviewer #2: Yes

5. Review Comments to the Author

Reviewer #1: This paper provides a prediction of bicycle route choice problem, and which develops a data-driven bicycle route choice model using GPS tracks from cyclists and uses DBSCAN to cluster the GPS tracks and group similar tracks. Then this paper trains a LSTM in order to find the cluster to which the GPS track belongs from its shortest path. The findings provide some insights while some contents need to be revised and improved.

1) The authors should list the specific contribution of this work.

2) I suggest that the limitation of this work should be discussed.

3) Page 1, Abstract: It is unclear to show the overall idea and purpose of this paper.

4) Page 1, Introduction: The distribution of this part is unreasonable, and the introduction of LSTM is relatively less. The content of literature review is few. Some references provided are outdated. And the research question and gap are not clear.

5) Page 4, in section 3.1: “Two data-sets are used.” However, there is no specific description of the two types of data collection events and processes. It is necessary to add the example of these two data-sets and give the standard of screening data.

6) Page 4, in section 3.2: “We noted that 10% of the experienced tracks are shorter than the generated shortest path.” However, no pictures or other source to support this conclusion.

7) Page 4, in section 3.3: First sentence, “In order to verify that cyclists do not use the shortest paths.” In the end of this paragraph, “This is consistent with our hypothesis: cyclists would use the shortest route to reach preferred roads in the city center.” Is there any contradiction?

8) Page 5, in the part of quantitative comparison: “Fig 3 shows the maximum detour reaches almost 100km.” It is not shown in Fig 3 that the maximum detour reaches almost 100km. Also, it is not clear to see from the graph that the 10% of experienced tracks shorter than their computed tracks.

9) Page 8, in section 5.1: The cyclability coefficient takes values in [0,1], not ]0,1].

10) Page 9, the Algorithm 1: The authors should give some explanation for each symbol and process of the algorithm.

11) The authors should to pay attention to English grammar, spelling, and sentence structure so that the goals and results of the study are clear to the reader. (Extra words appear in some sentences: Tracks created by our model are way more similar to the original GPS tracks…)

Reviewer #2: What are the practical contributions of this study?

There are already many machine learning methods, why use the DBSCAN and LSTM models?

Some dynamic factors for predicting the bike routes are not considered, for example, the weather condition, ect.

The proposed bike route choice model should be compared with the existing models in terms of the prediction accuracy.

Some recent studies should reviewed, for example, see: On the development of a semi-nonparametric generalized multinomial logit model for travel-related choices. PloS one, 12(10), e0186689. Radial Basis Function Neural Network with Localized Stochastic-Sensitive Autoencoder for Home-Based Activity Recognition. Sensors, 20(5), 1479.

6. PLOS authors have the option to publish the peer review history of their article (what does this mean?). If published, this will include your full peer review and any attached files.

Reviewer #1: No

Reviewer #2: No

---

## [Author Response · Author response to Decision Letter 0]

29 Nov 2021

We have answered all the reviewers' requests in the "Response to reviewers" included in the files. We thank the reviewers again for allowing us to improve our paper through their remarks.

---

## [Decision Letter · Decision Letter 1]

7 Feb 2022

Implicit GPS-based bicycle route choice model using clustering methods and a LSTM network

PONE-D-21-18342R1

Dear Dr. Magnana,

We’re pleased to inform you that your manuscript has been judged scientifically suitable for publication and will be formally accepted for publication once it meets all outstanding technical requirements.

Kind regards,

Jinjun Tang

Academic Editor

PLOS ONE

Reviewers' comments:

Reviewer's Responses to Questions

**Comments to the Author**

1. If the authors have adequately addressed your comments raised in a previous round of review and you feel that this manuscript is now acceptable for publication, you may indicate that here to bypass the “Comments to the Author” section, enter your conflict of interest statement in the “Confidential to Editor” section, and submit your "Accept" recommendation.

Reviewer #2: (No Response)

2. Is the manuscript technically sound, and do the data support the conclusions?

Reviewer #2: (No Response)

3. Has the statistical analysis been performed appropriately and rigorously? 

Reviewer #2: (No Response)

4. Have the authors made all data underlying the findings in their manuscript fully available?

Reviewer #2: (No Response)

5. Is the manuscript presented in an intelligible fashion and written in standard English?

Reviewer #2: (No Response)

6. Review Comments to the Author

Reviewer #2: (No Response)

7. PLOS authors have the option to publish the peer review history of their article (what does this mean?). If published, this will include your full peer review and any attached files.

Reviewer #2: No

---

## [Editor Report · Acceptance letter]

8 Mar 2022

PONE-D-21-18342R1 

Implicit GPS-based bicycle route choice model using clustering methods and a LSTM network 

Dear Dr. Magnana:

I'm pleased to inform you that your manuscript has been deemed suitable for publication in PLOS ONE. Congratulations! Your manuscript is now with our production department. 

Kind regards, 

on behalf of

Dr. Jinjun Tang 

Academic Editor

PLOS ONE